

**A retrospective study of the pre-eruptive unrest on El Hierro (Canary Islands): implications of seismicity and deformation in the short-term volcanic hazard assessment**

Stefania Bartolini[1,a], Carmen López[2], Laura Becerril[1], Rosa Sobradelo[3], Joan Martí[1]

[1] Group of Volcanology, (SIMGEO-UB) CSIC, Institute of Earth Sciences Jaume Almera, c/Lluis Sole Sabaris s/n, 08028 Barcelona, Spain.

[2] Observatorio Geofísico Central, Instituto Geográfico Nacional (IGN), c/Alfonso XII, 3, 28014 Madrid, Spain.

[3] Willis Research Network and Analytics Technology, Willis Towers Watson, London, UK.

[a] Corresponding author: Stefania Bartolini, Group of Volcanology, (SIMGEO-UB) CSIC, Institute of Earth Sciences Jaume Almera, c/Lluis Sole Sabaris s/n, 08028 Barcelona, Spain. (sbartolini.1984@gmail.com)

**Key points**

· Short-term spatio-temporal analysis for understanding unrest indicators during an unrest phase.

· A new methodology to be applied in short-term hazard assessment.

· Spatio-temporal analysis using information obtained from monitoring data.



**Abstract**
The correct identification and interpretation of unrest indicators are useful for
forecasting volcanic eruptions, delivering early warnings, and understanding the
changes occurring in a volcanic system prior to an eruption. Such indicators play an
important role in upgrading previous long-term volcanic hazard assessments and help
grasp the complexities of the preceding period of eruptive activity. In this work, we
present a retrospective analysis of the 2011 unrest episode on the island of El Hierro
that preceded a submarine eruption. We use seismic and surface deformation
monitoring data to compute the susceptibility analysis (QVAST tool) and to study the
evolution over time of the unrest (ST-HASSET tool). Additionally, we show the
advantages to be gained by using continuous monitoring data and hazard assessment e-
tools to upgrade spatio-temporal analyses and thus visualize more simply the
development of the volcanic activity.
**Keywords**
Short-term volcanic hazard assessment, unrest, precursors, monitoring, spatio-temporal
analysis




## 1. Introduction

The most challenging aspect of forecasting volcanic eruptions is the correct identification and interpretation of precursors during the episodes of unrest that normally precede eruptive activity. During this phase, the short-term volcanic hazard assessment can be computed by combining a long-term hazard analysis with real-time monitoring data, updating continuously the status of the volcanic hazard (Blong, 2000; Sobradelo and Martí, 2015; Tonini et al., 2016). Short-term evaluations can help forecast the likely outcomes – i.e. where and when the eruption will take place – by drawing on the information derived from indicators and an understanding of the volcanic system. The parameters associated with the volcanic process are the geophysical and geochemical signals that provide information on magma movement within the volcanic system and on how the magma is preparing to reach the surface (Chouet, 1996; McNutt, 1996).

In particular, the signals recorded during unrest episodes – for example, an increase in activity compared to the previous background level (Phillipson et al., 2013) – can be used to deduce changes in magma accumulation and movement, the state of stress of the host rock, and the physical and chemical properties of the magma itself (Harrington and Brodsky, 2007; Jellinek and Bercovici, 2011; Lavallée et al., 2008; McNutt, 2005; Neuberg et al., 2000; Papale, 1999; Tárraga et al., 2014). A comprehensive well organized monitoring network on and around the volcano is fundamental if scientists are to analyze how the eruption process is evolving. Changes may be detected on the surface that reflect variations in the geophysical (e.g. seismicity, surface deformation, and changes in potential fields) and/or geochemical (e.g. gas flow rate and gas composition) parameters sensed by the network that is monitoring the



activity of the volcano (Scarpa and Tilling, 1996; Sparks, 2003; Vallianatos et al., 2013,
Telesca et al., 2015).

It is essential that all the monitoring information obtained during an unrest phase

be processed and interpreted in real time. This is a crucial consideration since this
information is vital in eruption forecasting and provides support for decision-makers. In
many instances during an unrest phase, the institution in charge of the monitoring
network is expected to publish daily or even hourly bulletins with updates derived from
monitoring signals. These bulletins are then used by experts (e.g. a scientific committee
or crisis team) to keep public officials abreast of the state of the volcanic system. These
reports do not generally contain probabilistic model results and tend to consist merely
of processed monitoring data related to seismicity, deformation, and gas emissions.

In order to provide a simple and automated way of assessing the evolution of the

volcanic system from looking at the monitoring signals, the ST-HASSET was
developed (Sobradelo and Martí, 2015; Bartolini et al., 2016). This e-tool offers an
alternative to the BET-EF (Marzocchi et al., 2008) and BET-UNREST (Tonini et al.,
2016) and also proposes a flexible probabilistic approach to incorporate monitoring
information for the quantification of short-term volcanic hazard that looks for
significant changes in the values of the measured unrest indicators, across consecutive
time intervals. In comparison to the BET-EF and BET-UNREST, ST-HASSET does not
focus on the absolute value of each variable with respect to a defined threshold, but
compares its degree of change with respect to the previous time interval. In each case, a
variation that is considered significant can be defined in advance given the specific
characteristics of the volcano being studied.

Assuming that the geophysical indicators such as seismicity and ground

deformation provide insights on the location of magma during the unrest phase (Endo



and Murray, 1991; Chouet, 1996; McNutt, 1996; Martí et el., 2013), changes in the
location of such unrest parameters may indicate magma movement and, consequently,
that the location of potential new vents may also change. This is extremely important
when conducting hazard assessment analysis, as the location of the eruptive vent may
condition the resulting hazards and their potential impacts. In this sense, short-term
hazard assessment needs to inform in real time on how monitoring information changes
the probabilities of vent opening (volcanic susceptibility) and of the hazards that may
occur, as well as of the proximity of the eruptive event.

In order to show how ST-HASSET works, we apply it retrospectively to the

unrest episode that preceded the El Hierro eruption in 2011. When volcanic unrest
started here in July 2011, the Spanish National Geographical Institute (IGN), the
institution responsible for volcano monitoring in Spain, set up a dense seismic
monitoring network composed of a three-component (3CC) broadband station (CTIG)
and eight short- and medium-period (natural periods of 1 and 5 s) 3CC stations (López
et al., 2012) (Fig. 1). In order to monitor the associated 3D deformation, the IGN also
deployed four extra GPS stations on El Hierro to reinforce the capacity of the single
pre-existing GPS station (FRON) (Fig. 1) belonging to the Canarian Regional
government (López et al., 2012, 2014; Martí et al., 2013). The amount of information
registered provides a good example of a monitored unrest episode with a complete
dataset. However, during the pre-eruptive unrest phase the continuous changes in the
position of the seismicity and deformation sources made it all but impossible to forecast
the position of the new vent and, consequently, to define reliable eruption scenarios.

The objective of this retrospective analysis is to define guidelines on how we

can manage the information generated by a monitoring network during the unrest phase
of an ongoing crisis. We use the data recorded in the pre-eruptive unrest episode that

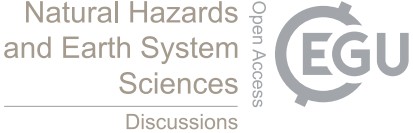


100 took place on El Hierro in 2011 to update in real time the spatial probability of the new

101 vent opening and to interpret the unrest precursors as a means of determining the

102 probability of evolution of these indicators. So, we first evaluate the volcanic

103 susceptibility combining the real time monitoring information with the QVAST tool

104 (Bartolini et al., 2013), which provides a real time variation of the vent opening

105 probabilities. Then, we combine each updated result with the ST-HASSET tool to

106 determine the evolution over time of the unrest indicators. The results obtained allow us

107 to realise how the application of these tools helps interpret the unrest indicators and how

108 they can be used for improving the susceptibility assessment and the definition of

109 realistic eruptive scenarios, thus facilitating the decision making process and the

110 management of the volcanic crisis.

112 **2. Methodology**

113   The methodology used in this study basically consists in the systematic

114 application of two e-tools specifically designed for conducting probabilistic spatial and

115 temporal analysis in volcanic hazard assessment.

116   QVAST (Bartolini et al., 2013) is a tool that has been developed to evaluate the

117 spatial probability of a new vent opening (volcanic susceptibility) using volcano-

118 structural data and seismicity. In monogenetic volcanism, as it is the case of El Hierro,

119 each new eruption creates a different vent, which indicates that accurate spatial

120 forecasting is highly uncertain. This type of analysis has been often applied in long-term

121 hazard assessment as it represents a good starting point for developing hazard maps

122 based on certain assumptions: i) future eruptive vents will be close to the previous ones

123 and ii) the stress field plays the most significant role in determining where magma will

124 reach the surface (see Martí et al., 2016). The result is a (long-term) susceptibility map





obtained by assigning different weights to each of the probability density functions in
each dataset (volcano-structural elements: location of past vents, eruptive fissures,
fractures, faults, dykes, etc.) considered in the analysis, which are combined via a
weighted sum and modelled in a non-homogeneous Poisson process. During an unrest
phase, the (short-term) susceptibility map varies as new information (e.g. the location of
the seismic events) is provided by monitoring data. Hence, the previously defined
probabilities of hosting a new vent will change in terms of where the new seismicity
and/or ground deformation is located — assuming that both parameters provide an
indication of magma movement and location.

The probability of occurrence of a possible eruptive scenario will change

according to the variations in the short-term susceptibility map, which will be redefined
each time that new monitoring information will be computed; thus, we also have to
calculate the temporal evolution of monitoring data.

The ST-HASSET tool (Sobradelo and Martí, 2015; Bartolini et al., 2016) is a

simple tool that develops an event tree structure that uses a quantitative approach via
Bayesian inference to assess the hazard of a particular volcanic scenario by taking into
account monitoring data and all relevant data pertaining to the past history of the
volcano. Indicators are shown on a common probability scale to visualize their progress
during the unrest phase and to estimate the probability of occurrence of a particular
eruptive scenario.

**146     3. Unrest on El Hierro in 2011**

El Hierro, situated in the southwestern corner of the Canary archipelago (Fig. 1), is
geologically the youngest of these islands and its oldest subaerial rocks have been dated
at 1.12 Ma (Guillou et al., 1996). It corresponds to a shield structure formed by different



volcanic edifices with three rift zones along which recent volcanism has been
concentrated (Guillou et al., 1996). The studied unrest period started on 19 July 2011
and gave rise to a submarine eruption that started on 10 October 2011 (Fig. 1). The
whole episode was well monitored by the IGN and during the period leading up to the
eruption approximately 10,000 earthquakes with local magnitudes of up to 4.3 were
recorded, and over 5 cm of vertical and horizontal surface deformations were registered
(López et al., 2014).
This pre-eruptive unrest started with a marked increase in seismicity, followed a
few days later by surface deformation and gas emissions (López et al., 2012). The
evolution of the seismicity during this episode was characterized by changes in the
hypocentral location that were interpreted as movements in the position of the magma
(Fig. 2 and Table 1) (López et al., 2012, 2014). During the first weeks of unrest, all the
seismic events were located in the north of the island at a depth of about 10–15 km b.s.l.
and were of low magnitude. As of 4–26 September 2011, the seismicity migrated
southwards along the crust/mantle boundary and the amount of released seismic energy
increased. GPS stations began also to rotate to the North, suggesting a simultaneous
surface deformation pattern that reflected a correlated migration of the pressure source
towards the south. From 27 September to 7 October 2011, both the seismic rate and the
seismic energy grew and events were now located mostly off the SW coast of El Hierro.
At the same time, a sudden deflation–reinflation was observed on the N–S component at
all GPS stations (1–5 October 2011). On 8 October at 20:34 h (GMT), a 4.3 ML
earthquake (the greatest magnitude recorded during the unrest period) occurred 1.5 km
off the SW coast of the island at a depth of 12 km. However, from this moment
onwards, very few further earthquakes were registered and the pre-eruptive episode
culminated with a submarine eruption on the southern flank of the island's volcanic



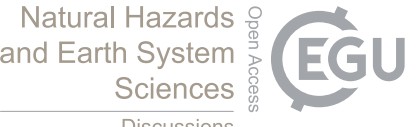

edifice (López et al., 2012) (Fig. 1). On 10 October at 04:10 UTC, a clear emergent
tremor signal was registered by all the seismic stations indicating the onset of the
eruptive activity that lasted for more than four months (until the end of February 2012)
(López et al., 2014).

**4. Datasets**
4.1 Spatial analysis
A susceptibility analysis enables us to determine the probability of occurrence of future
eruptive vents. This probability depends on the volcano-structural elements that define
the structural setting of a volcano and the past pathways taken by the magma as it
ascended to the Earth's surface. Eruptive vents and fissures, dykes, faults, fumaroles,
and the stress field are the most important elements (Martin et al., 2004; Jaquet et al.,
2008; Cappello et al., 2012; Bartolini et al., 2013; and references therein) that determine
the probabilities of an eruptive vent opening in an area that was affected by similar
types of eruptions in the past.

In order to compute the probability of opening a new eruptive vent at El Hierro,

we took into account the most relevant volcano-structural data as given by Becerril et al.
(2013, 2014) (Fig. 2): (i) the subaerial vents and eruptive fissures that are part of the
Rift Volcanism (including sub-recent and recent eruptions) and (ii) the submarine vents
and eruptive fissures deduced from bathymetric inference. Furthermore, we chose only
those eruptive fissures oriented between N00ºE and N45ºE in relation to the orientation
of the regional stress field (see Geyer et al., 2016). We assumed that the stress field
plays the most important role in determining where the magma will reach the surface
and the fractures orientated in this direction were those that offered the least resistance
to magma transport.




To conduct the short-term analysis, we complemented the previous dataset with
the addition of data on the evolution of the seismicity for the unrest period (19 July
2011–10 October 2011).
We assumed that in this short-term spatial analysis the location of the seismicity
reflected the position of the magma, as it provides a good indicator for tracking magma
migration and for determining where it may potentially reach the surface. However, the
location of gas emission was not considered in this short term analysis as they were too
disperse in the whole area (López et al., 2012) and thus not sufficiently informative on
the position that magma could have below the island. Concerning the surface
deformation, we considered this parameter only in the temporal analysis, due to the lack
of a well-distributed ground deformation monitoring network operating during the El
Hierro unrest episode. So, as described in López et al. (2012), the highest values of
uplift were found in the area where the seismicity moved from north to south and where
no GPS was available.
Seismic data was obtained from the seismic catalogue published by the Spanish
National Geographical Institute (IGN) (www.ign.es) (Fig. 2). Data was grouped in time
windows of four days to optimize the forecast given that certain volcanic systems have
indicated that magmatic processes have a memory with a time-scale of just a few days.
(Connor et al., 2003; Jaquet and Carniel, 2003; Jaquet et al., 2006; Tárraga et al., 2006;
Carniel et al., 2006). Such a selection allows assuring the persistent behaviors of the
system. Within the time window, the seismic activity will follow the same trend of
previous days, allowing the short-term forecast. We selected from the IGN catalogue
only those earthquakes of magnitudes greater than zero and precise locations, with
epicenter maximum semi-ellipse axes of less than 15 km, minimum semi-ellipse axes of
less than 6 km, and a depth error of less than 8 km. In this way, we aimed to avoid –


inasmuch as was possible – errors in the hypocenter localization of earthquakes due to
the small number of the seismic stations in place during the first unrest phase.

4.2 Temporal analysis
The data for the temporal analysis consisted of observables which relative variation
with time may indicate changes in the processes occurring inside the volcano when
preparing for a new eruption (Sobradelo and Martí, 2015; Bartolini et al., 2016). In our
methodology, we do not use the absolute values of each parameter, but considering their
relative variation with time, we only indicate if there is an increase or a decrease in the
value of such parameter in each time interval. We used the monitoring data gathered by
the IGN and other published information (López et al., 2012, 2014; Martí et al., 2013;
Telesca et al., 2014). This information is given in Table 1 and includes:
-   the number of seismic events
-   RSAM (Real-time Seismic Amplitude Measurement)
-   the seismic energy released during the fracturing process
-   the lateral and vertical migration of the seismicity
-   the number of shallow seismic events
-   the strain variation.

Therefore, consistent with the choices adopted for the spatial analysis, the

variation in the unrest indicators (increase/decrease) was evaluated in relation to the
mean values for the previous four days. The seismic rate variation was considered by
taking into account only those events with a magnitude over 2.5 (greater than the
completeness magnitude during almost all the period), assuring this way the study of the
seismic evolution (López et al., 2017), while a significant change was considered only
when the rate of variation was 25% higher in relation to the previous four days as a



consequence of stress reorganization (Stein, 1999). RSAM data was obtained by
analyzing the signal registered by the vertical component of the seismic broadband
CTIG station (Fig. 1). Although the signal may have a high background of seismic
noise, a RSAM increase is a good indicator of the transport of the magma to the surface
(Endo and Murray, 1991). In order to highlight a significant increase in RSAM values,
we considered the variation in the slope of the inverse of the RSAM, which is clear
evidence of a consistent increase in the signal. The accumulated increase in energy
release was considered to be significant when the energy value (i.e. the accumulated
value in relation to the mean value over the last four days) was greater than 10%. In this
case, the accumulated energy curve showed a notable slope variation. For the lateral
migration of the seismicity, we considered a significant variation to exist when the
displacement increment was over 1 km. This is compatible with the effects on hazard
scenarios when the vent location changes. The vertical migration of the seismicity
ranges from a depth of approximately 19 km to the surface and, taking into account the
mean of the variation, a variation greater than 0.6 km was assumed to be notable. The
number of shallow events reflects the presence of the magma close to the surface and so
we assumed that the number of events of magnitude greater than 3 in the same day at a
depth of 0–5 km was significant. Finally, the strain variation has been determined with
the horizontal components data of the GPS FRON station. We have assumed a
significant variation when the increase/decrease was greater than 1.5 mm of the vector
representing the horizontal deformation (composing the north and east GPS
components).


**5. Results**





### 5.1 Spatial probability of new vent opening

Given its great flexibility and ability to identify the most likely zones to host new eruptions in monogenetic volcanic fields, we used the QVAST tool (Bartolini et al., 2013) to determine the susceptibility from the evolution of the seismicity during the unrest. This tool was applied first to evaluate the smoothing factors (bandwidths) of the dataset analysed, then to evaluate the probability density functions for each dataset, and, finally, to calculate the final susceptibility map (Fig. 3) (see also Figure S1).

In the case of the rift volcanism and the submarine layers, we applied the Least Square Cross Validation Method (LSCV) (Cappello et al., 2012; Bartolini et al., 2013) to obtain the bandwidth parameter (see Becerril et al. (2013)). To determine the influence of seismicity in the spatial analysis, we considered that the most representative result was that obtained using Silverman's Rule of Thumb for the optimal bandwidth (Silverman, 1986). Thus, we obtained a bandwidth value of 1100 m for the rift volcanism and of 3900 m for the submarine layer, while in the case of the seismic data the range in the degree of randomness was from 500 m to 1500 m.

In the evaluation of the final susceptibility, weights were assigned based on expert opinion and on previously published work (Becerril et al., 2013, 2014), and by taking into account the average depth of the seismicity during the unrest episode. Specifically, up to 7 October we observed no significant variation in the shallow seismicity (Table 1). In this case, we assigned the following weights: 0.5 for seismic events, 0.3 for onshore vents and fissures, and 0.2 for offshore vents and fissures. In the final period (8–10 October), we considered the shallow earthquakes as a separate layer by assigning a different and more consistent weight as follows: 0.6 for shallow seismic events, 0.2 for the remaining seismic events, 0.1 for onshore vents and fissures, and 0.1 for offshore vents and fissures.


The results shown in Figure 3 (see also Figure S1) highlight the importance of

combining monitoring data with a previous long-term hazard assessment as a means of

updating the probability of a new eruptive vent appearing in a particular area. The

presence of previous volcanic structures does not provide sufficient information for

forecasting the possible opening of a fresh vent during the unrest phase; however, if this

information is combined with ongoing seismicity the predicted result can be improved.

As shown in Figure 3, before the eruption the area with the highest probability of a

fresh vent opening is the area that is closest to the eruptive vent.

**309       5.2 Evolution of unrest indicators and short-term hazard assessment**

The temporal analysis of the unrest indicators was conducted by applying the ST-

HASSET tool (Bartolini et al., 2016) to analyze possible patterns in the evolution of

events preceding the submarine eruption on El Hierro. The advantages of this tool lie in

its ability to consider different signals on the same probabilistic scale, based on any

significant or abnormal change in the unrest signal, with respect to a previous stage

and/or a base-line measurement considered normal. The tool computes at each stage the

probability of experiencing an anomalous change (increase/decrease) by the next time

bulletin, based on what has been observed up until now. With this, it helps the scientist

sum up the evolution of the unrest indicators and gets some insight into the possible

unfolding of the volcanic crisis in the immediate future, helping with decision-making

and the interpretation of the unrest. In Table 1, we show the data for the entire unrest

period and, as explained in section 3.2.2, we considered the variation ("Y") of the

indicator analysed based on different criteria. The choice of the aleatoric and epistemic

uncertainties (prior and data weigths) surrounding the probability estimates were

assumed considering that El Hierro unrest was the first unrest registered in Canaries.





The prior weights were assumed to be the probability results of the previous bulletin
(only in the first simulation we have assumed the same probability for each indicator).
In the case of the data weights, we have first assigned a total epistemic uncertainty and
sequentially incremented the weight with the evolution of the unrest.

In Figure 4 and Movie S1, the evolution of the indicators over the entire unrest

period with a daily time window are clearly visible. In the right side of the chart, it is
shown also day-by-day the total number of parameters that increase or descrease during
the unrest evolution. We assumed a value of +1 if the indicator increases, -1 if the
indicator decreases, and 0 if the change is not significant. This allows visualizing the
overall tendency of variation of the unrest indicators. We also considered three phases
of 28 days, all three during the evolution of the unrest period, as shown in Figure 5 and
so were able to observe how these indicators varied in different ways as the unrest
evolved:

– Phase I: from July 19th to August 15th;

– Phase II: from the August 16th to September 12th;

– Phase III: from September 13th to October 10th.

By having all the precursory activity mapped and plotted into the same graph, it is
easier to interpret their evolution as a whole. According to what was been defined as a
significant change, in a first phase the accumulated energy released increase (AERI) and
the lateral migration of seismicity (LMS) experienced a significant change, and
continued overall the increasing tendency across this initial phase with periods of no
significant variation followed by periods of heavy changes. By the time, they enter the
second phase both indicators show no changes seem stable until well into the third
phase where AERI starts experiencing significant increases and LMS follows a few
days later. As per the other indicators, in a first phase they all experience a significant



change at some point in the initial stages of Phase I and seem to enter a quiet phase after
that, except for the RSAM, which on average experiences a continuous increase across
the three phases, perhaps more consistent though Phases II and III. The unrest indicators
that seem to experience larger significant changes in Phase I are AERI, LMS and
RSAM.
Phase II was characterized by an overall stabilization of the indicators, except
for RSAM that continues to consistently increase. In addition, by the middle of this
second phase the seismicity experiences a significant increase with a small period of
significant lateral migration of seismicity, followed by a small jump in the RSAM a few
days later. By the time the systems enters into the phase III on September 12 we
continue to observe a probability increase in RSAM with a new significant jump around
the September 18. This change happens simultaneously with a significant LMS increase
for the first time since phase I, and a jump in the seismicity increase followed by an
AERI jump and strain variation. There seems to be a clear inflection point around the
20th of September where all unrest indicators at once, for the first time since the
beginning of the unrest three months ago, begin to show consistently significant
changes, indicating the system has changed and is getting ready to enter into a new
eruptive phase. Note that a few days before the submarine eruption there is a jump in all
the indicators including for the first time the shallow seismicity and the vertical
migration of seismicity, the probabilities for these two continue to increase from this
moment onwards, together with RSAM, while LMS and AERI remain constant.

**6. Discussion and Conclusions**
Short-term hazard assessment should be always conducted based on a previous long-
term hazard assessment, as a systematic study of past eruptive activity conducted well





before a new volcanic crisis starts can help forecast the most probable scenarios and
thus avoid confusion regarding the potential outcome of the forthcoming eruption.
In the case of El Hierro, unfortunately, no previous hazard assessment existed, so the
most probable scenario – a submarine eruption – was not anticipated, as has been shown
by a subsequent study (Becerril et al., 2014). Consequently, scientific advisors and
decision-makers considered possible eruptive scenarios that had much lower
probabilities of occurrence, which implied the taking of decisions with a higher cost
than necessary (Sobradelo et al., 2014).

Via a retrospective analysis of the particular case of El Hierro, the results

obtained in this work provide an easy and useful approach to the understanding and
visualization of the information recorded by the monitoring system, and show how this
information can be used to forecast an eruption and its potential hazards in real time.
The translation of this information into a coherent picture that will be helpful for
anticipating the future evolution of a volcanic system is not straightforward, which is
why we propose that this simple methodology be used to facilitate communication
among scientists and between scientists and decision-makers. Moreover, the
interpretation of unrest indicators and the observation of significant variations in
volcanic systems are complex tasks subject to great uncertainties and the approach
proposed in this work aims to act as a guide for experts and decision-makers to be
employed as a crisis unfolds.

Another important aspect is how to interpret monitoring signals in monogenetic

volcanism. In this specific case, where the location of a future eruption is not easy to
determine, the spatial probability is controlled by local and regional stress fields that are
usually poorly understood. During the pre-eruptive episode on El Hierro, it was clear
that the lateral migration of the magma was controlled by the presence of stress barriers





defined by major structural and rheological discontinuities (Martí et al. 2013, 2017).
This gave rise to nearly continuous changes in the probable location of the eruptive
vent, which hindered the definition of a precise eruptive scenario and the application of
appropriate mitigation measures. This highlights the importance of understanding
monitoring signals and their interactions, as well as the need for knowledge of the past
activity of the volcanic system in the form of susceptibility and hazard analyses, if a
volcanic eruption is to be correctly forecast. In case of El Hierro, the susceptibility map
that combines volcano-structural information and seismic data (Fig. 3) shows how the
possible location of a eruptive vent varied during the evolution of the pre-eruptive
unrest: initially, the magma was thought to be accumulating on the northern side of the
island (Fig. 3b) but in the end it was concentrated on the southern side (Fig. 3d), where
it eventually provoked a submarine eruption. This confirms the idea that seismic activity
and ground deformation are good indicators of magma location in monogenetic
volcanism.

The analysis of the precursors shows how special attention should be paid to

each one during the evolution of the unrest period (Fig. 4). Indeed, in the initial phase,
we observed obvious fluctuations in most indicators and, above all, an increase in the
accumulated energy released compared to the background level. In the second phase,
the behavior of these indicators remained constant and there was no significant spread, a
reflection of how the magma followed the local stress field and migrated from the north
to the southeast. During the final month before the eruption, we noted that the indicators
started to increase sequentially but at the same hypocentral depth. However, in the final
hours before the eruption the presence of very shallow seismicity indicated that,
immediately after the final major earthquake, a relatively rapid vertical migration of
magma was taking place. This vertical ascent to the surface was associated with a





drastic decrease in both the number of seismic events (almost no seismicity of any kind
in the 30 hours before the onset of the eruption) in the accumulated energy release, and
in the deformation, but also with an increase in the RSAM, thereby suggesting that the
final major tectonic earthquake facilitated a path for the magma to reach the surface
(Martí et al., 2013).

From an emergency management perspective, it is worth stressing two further

important results of the application of our method. Firstly, it identified unmistakably the
anomalous behavior of the activity, characterized by an increasing probability in almost
all indicators during the first days of the unrest period as they varied in relation to the
background values. Secondly, many indications suggested that the probability of an
eruption increased in almost all parameters from 25 September until the onset of the
eruption. On 23–27 September, the Canarian Civil Protection Authorities in charge of
the management of the volcanic crisis changed the alert level for the population from
Green to Yellow in two areas due to the strong seismicity being felt by the population
and the risk of rock falls near populated areas. In 11 October, the appearance of an
increasingly strong seismic tremor signal in the monitoring network warned of the
imminent onset of the eruption and Civil Protection raised the alert level to Red.
Despite the correct management of the eruption crisis on El Hierro by the Canarian
Civil Protection, we still believe that our results can improve significantly the island's
early warning capability during an unrest period characterized by a high level of
uncertainty. Thus, the tools presented here could have been very useful for the Canarian
Civil Protection during the October 2011 eruption crisis.

**Acknowledgements:** This research was funded by the European Commission (EC
ECHO Grant SI2.695524: VeTOOLS). The authors would like to thank the Instituto



Geográfico Nacional (IGN - Madrid) and, especially, Rafael Abella Meléndez, for the
ease in accessing the monitoring data. Joan Martí is grateful for the MECD
(PRX16/00056) grant. The English text was edited by Michael Lockwood.

























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





**Table**
Table 1. Unrest indicators during the unrest period.

**Figures**
Figure 1. Location of El Hierro and the IGN monitoring network during the unrest
period.
Figure 2. Structural data of El Hierro and the evolution of the seismicity during the
unrest period (average location of the seismic swarm).
Figure 3. Susceptibility maps obtained from: a) the volcano-structural data; b) the first
days of unrest; c) in the middle of the unrest; d) the days before the submarine eruption.
Figure 4. ST-HASSET: the evolution of the unrest indicators in three phases of 28 days.
The right side of the chart shows day-by-day the tendency of variation of the unrest
indicators.
Figure 5. ST HASSET: the evolution of all indicators every 28 days (3 phases of
unrest).

**Supplementary Material**
Figure S1 - Susceptibility maps and seismicity location during the evolution of the
unrest period.
Movie S1 - Evolution of the unrest indicators and its registered values.


Table 1

| UNREST INDICATORS | SEISMICITY INCREASE | | | RSAM ACCELERATION INCREASE | | | ACCUMULATED ENERGY RELEASED INCREASE | | | LATERAL MIGRATION OF SEISMICITY | | | VERTICAL MIGRATION OF SEISMICITY | | | SHALLOW SEISMICITY | | | STRAIN VARIATION | | |
|---|---|---|---|---|---|---|---|---|---|---|---|---|---|---|---|---|---|---|---|---|---|
| | Y/N/na | Value [n°] | Probability | Y/N/na | Value [RSAM unit] | Probability | Y/N/na | Value [J] | Probability | Y/N/na | (Figure 2) | Probability | Y/N/na | Value [km] | Probability | Y/N/na | Value [n°] | Probability | Y/N/na | Value [m] | Probability |
| 2011-07-19 | N | 0 | 0.333 | N | 24.62 | 0.333 | N | 5.80E+07 | 0.333 | N | | 0.333 | N | 12 | 0.333 | N | | 0.333 | N | 0.012 | 0.333 |
| 2011-07-20 | N | 0 | 0.25 | N | 23.80 | 0.250 | Y | 2.41E+08 | 0.500 | Y | | 0.500 | N | 11.88 | 0.250 | N | 0 | 0.25 | N | 0.014 | 0.25 |
| 2011-07-21 | N | 0 | 0.2 | N | 25.38 | 0.200 | Y | 1.72E+09 | 0.600 | N | | 0.400 | N | 10.78 | 0.200 | N | 0 | 0.2 | N | 0.015 | 0.2 |
| 2011-07-22 | N | 0 | 0.167 | N | 23.34 | 0.167 | N | 3.18E+09 | 0.667 | N | | 0.333 | N | 10.09 | 0.167 | N | 0 | 0.167 | N | 0.015 | 0.167 |
| 2011-07-23 | N | 0 | 0.143 | N | 21.21 | 0.143 | Y | 2.15E+09 | 0.715 | Y | | 0.428 | N | 10.72 | 0.143 | N | 0 | 0.143 | N | 0.014 | 0.143 |
| 2011-07-24 | N | 0 | 0.125 | N | 19.55 | 0.125 | Y | 4.30E+08 | 0.751 | Y | | 0.499 | N | 10.53 | 0.125 | N | 0 | 0.125 | N | 0.012 | 0.125 |
| 2011-07-25 | N | 0 | 0.111 | N | 18.66 | 0.111 | Y | 5.73E+08 | 0.779 | Y | | 0.555 | Y | 13.4 | 0.222 | N | 0 | 0.111 | N | 0.014 | 0.111 |
| 2011-07-26 | N | 0 | 0.1 | N | 19.02 | 0.100 | Y | 1.50E+09 | 0.801 | N | | 0.499 | N | 8.45 | 0.200 | N | 1 | 0.1 | N | 0.014 | 0.1 |
| 2011-07-27 | Y | 2 | 0.182 | N | 19.19 | 0.091 | Y | 3.00E+09 | 0.819 | N | | 0.454 | N | 8.93 | 0.182 | Y | 4 | 0.182 | N | 0.015 | 0.091 |
| 2011-07-28 | N | 0 | 0.167 | N | 18.26 | 0.167 | Y | 7.27E+08 | 0.834 | N | | 0.416 | N | 10.33 | 0.167 | N | 1 | 0.167 | N | 0.016 | 0.083 |
| 2011-07-29 | N | 0 | 0.154 | Y | 22.92 | 0.231 | N | 1.79E+08 | 0.770 | N | | 0.384 | N | 12.75 | 0.154 | N | 0 | 0.154 | N | 0.016 | 0.077 |
| 2011-07-30 | N | 0 | 0.143 | Y | 27.70 | 0.286 | N | 6.17E+08 | 0.715 | N | | 0.357 | N | 11.82 | 0.214 | N | 0 | 0.143 | N | 0.018 | 0.071 |
| 2011-07-31 | N | 0 | 0.133 | Y | 28.73 | 0.267 | N | 1.22E+09 | 0.667 | N | | 0.333 | N | 11.18 | 0.200 | N | 0 | 0.133 | na | na | 0.071 |
| 2011-08-01 | N | 0 | 0.125 | N | 25.78 | 0.250 | N | 1.75E+08 | 0.625 | N | | 0.312 | N | 12 | 0.188 | N | 0 | 0.125 | N | 0.020 | 0.067 |
| 2011-08-02 | N | 0 | 0.118 | N | 21.44 | 0.235 | Y | 4.60E+08 | 0.588 | Y | | 0.352 | N | 10.67 | 0.177 | N | 0 | 0.118 | N | 0.017 | 0.063 |
| 2011-08-03 | N | 0 | 0.111 | N | 22.17 | 0.222 | N | 4.48E+08 | 0.555 | Y | | 0.388 | N | 11.47 | 0.167 | N | 0 | 0.111 | N | 0.020 | 0.059 |
| 2011-08-04 | N | 0 | 0.105 | N | 23.72 | 0.210 | Y | 2.60E+09 | 0.578 | Y | | 0.420 | N | 10.54 | 0.158 | N | 0 | 0.105 | N | 0.021 | 0.056 |
| 2011-08-05 | N | 0 | 0.1 | N | 26.11 | 0.199 | N | 4.24E+08 | 0.599 | Y | | 0.449 | N | 10.54 | 0.150 | N | 1 | 0.1 | N | 0.020 | 0.053 |
| 2011-08-06 | N | 0 | 0.095 | N | 24.29 | 0.190 | Y | 2.31E+08 | 0.618 | Y | | 0.475 | N | 9.61 | 0.143 | N | 0 | 0.095 | Y | 0.022 | 0.05 |
| 2011-08-07 | N | 0 | 0.091 | N | 17.88 | 0.181 | Y | 1.62E+09 | 0.635 | N | | 0.453 | N | 10.45 | 0.136 | N | 3 | 0.091 | Y | 0.027 | 0.093 |
| 2011-08-08 | N | 0 | 0.087 | Y | 15.15 | 0.217 | N | 1.46E+09 | 0.607 | N | | 0.433 | N | 11.16 | 0.130 | N | 1 | 0.087 | N | 0.022 | 0.089 |
| 2011-08-09 | Y | 1 | 0.125 | Y | 18.75 | 0.250 | Y | 4.38E+09 | 0.623 | N | | 0.415 | N | 10.6 | 0.125 | N | 2 | 0.083 | N | 0.021 | 0.085 |
| 2011-08-10 | N | 0 | 0.12 | Y | 26.84 | 0.280 | N | 1.58E+09 | 0.638 | N | | 0.398 | N | 10.95 | 0.120 | N | 0 | 0.08 | N | 0.024 | 0.082 |
| 2011-08-11 | N | 0 | 0.115 | N | 28.53 | 0.269 | N | 4.12E+08 | 0.613 | N | | 0.383 | N | 10.07 | 0.115 | N | 0 | 0.077 | N | 0.023 | 0.079 |
| 2011-08-12 | N | 0 | 0.111 | N | 26.71 | 0.259 | N | 2.72E+08 | 0.590 | N | | 0.369 | N | 11.28 | 0.111 | N | 0 | 0.074 | N | 0.022 | 0.076 |
| 2011-08-13 | N | 0 | 0.107 | N | 27.61 | 0.250 | N | 6.22E+07 | 0.569 | N | | 0.392 | N | 10 | 0.107 | N | 0 | 0.071 | N | 0.021 | 0.073 |
| 2011-08-14 | N | 0 | 0.103 | N | 28.80 | 0.241 | N | 1.40E+09 | 0.549 | Y | | 0.413 | N | 11.91 | 0.103 | N | 0 | 0.069 | N | 0.021 | 0.07 |
| 2011-08-15 | N | 0 | 0.1 | N | 26.39 | 0.233 | N | 6.61E+08 | 0.531 | N | | 0.399 | N | 11.41 | 0.100 | N | 0 | 0.067 | N | 0.024 | 0.068 |
| 2011-08-16 | N | 0 | 0.097 | N | 24.65 | 0.225 | Y | 1.73E+08 | 0.514 | Y | | 0.418 | N | 11.47 | 0.097 | N | 0 | 0.065 | N | 0.023 | 0.066 |
| 2011-08-17 | N | 0 | 0.094 | N | 28.33 | 0.218 | N | 6.48E+07 | 0.498 | Y | | 0.436 | N | 10.9 | 0.094 | N | 0 | 0.063 | N | 0.023 | 0.064 |
| 2011-08-18 | N | 0 | 0.091 | N | 22.92 | 0.211 | Y | 4.55E+09 | 0.483 | Y | | 0.453 | N | 10.39 | 0.091 | N | 2 | 0.061 | N | 0.023 | 0.062 |
| 2011-08-19 | N | 0 | 0.088 | N | 17.42 | 0.205 | Y | 2.16E+09 | 0.498 | Y | | 0.449 | N | 10.32 | 0.088 | N | 2 | 0.059 | N | 0.023 | 0.06 |
| 2011-08-20 | N | 0 | 0.085 | Y | 15.36 | 0.228 | N | 1.75E+08 | 0.484 | N | | 0.456 | N | 11.19 | 0.085 | N | 0 | 0.057 | N | 0.024 | 0.058 |
| 2011-08-21 | N | 0 | 0.083 | Y | 18.33 | 0.249 | Y | 2.25E+09 | 0.471 | N | | 0.443 | N | 11.12 | 0.083 | N | 0 | 0.055 | N | 0.025 | 0.056 |
| 2011-08-22 | N | 0 | 0.081 | Y | 22.89 | 0.269 | Y | 2.72E+09 | 0.458 | Y | | 0.458 | N | 11.12 | 0.081 | Y | 4 | 0.081 | N | 0.026 | 0.054 |
| 2011-08-23 | N | 0 | 0.079 | Y | 25.08 | 0.288 | N | 1.11E+09 | 0.446 | Y | | 0.472 | N | 10.97 | 0.079 | N | 0 | 0.079 | N | 0.026 | 0.053 |
| 2011-08-24 | N | 0 | 0.077 | N | 24.26 | 0.281 | N | 3.55E+07 | 0.435 | N | | 0.460 | N | 10.32 | 0.077 | N | 0 | 0.077 | N | 0.024 | 0.052 |
| 2011-08-25 | N | 0 | 0.075 | N | 19.57 | 0.274 | N | 2.55E+08 | 0.424 | N | | 0.449 | N | 10.75 | 0.075 | N | 0 | 0.075 | N | 0.025 | 0.051 |
| 2011-08-26 | N | 0 | 0.073 | N | 17.38 | 0.267 | N | 5.73E+07 | 0.414 | Y | | 0.462 | N | 10.9 | 0.073 | N | 0 | 0.073 | N | 0.026 | 0.05 |
| 2011-08-27 | N | 0 | 0.071 | Y | 16.03 | 0.284 | N | 1.17E+08 | 0.404 | N | | 0.451 | N | 11.23 | 0.071 | N | 0 | 0.071 | N | 0.025 | 0.049 |
| 2011-08-28 | N | 0 | 0.069 | Y | 17.48 | 0.301 | N | 2.56E+07 | 0.395 | N | | 0.441 | N | 10.95 | 0.069 | N | 0 | 0.069 | N | 0.028 | 0.048 |
| 2011-08-29 | N | 0 | 0.067 | Y | 24.46 | 0.317 | N | 2.42E+08 | 0.386 | N | | 0.431 | N | 10.69 | 0.067 | N | 0 | 0.067 | N | 0.027 | 0.047 |
| 2011-08-30 | N | 0 | 0.066 | N | 20.40 | 0.310 | N | 8.00E+08 | 0.377 | N | | 0.421 | N | 10.71 | 0.066 | N | 0 | 0.066 | N | 0.028 | 0.046 |
| 2011-08-31 | N | 0 | 0.065 | N | 11.87 | 0.303 | N | 7.45E+08 | 0.369 | N | | 0.412 | N | 11.61 | 0.065 | N | 1 | 0.065 | N | 0.027 | 0.045 |
| 2011-09-01 | Y | 1 | 0.085 | Y | 8.20 | 0.319 | Y | 4.35E+09 | 0.381 | N | | 0.403 | N | 11.2 | 0.064 | N | 0 | 0.064 | N | 0.028 | 0.044 |
| 2011-09-02 | Y | 2 | 0.104 | Y | 11.11 | 0.332 | Y | 3.96E+09 | 0.353 | N | | 0.395 | N | 11.02 | 0.063 | N | 1 | 0.063 | N | 0.031 | 0.043 |
| 2011-09-03 | N | 0 | 0.102 | Y | 16.66 | 0.346 | Y | 1.28E+09 | 0.346 | N | | 0.387 | N | 11.14 | 0.062 | N | 0 | 0.062 | N | 0.030 | 0.042 |
| 2011-09-04 | N | 0 | 0.1 | Y | 18.58 | 0.359 | N | 8.43E+08 | 0.339 | Y | | 0.399 | N | 10.79 | 0.061 | N | 0 | 0.061 | N | 0.030 | 0.041 |
| 2011-09-05 | N | 0 | 0.098 | N | 18.58 | 0.352 | N | 5.58E+08 | 0.332 | Y | | 0.411 | N | 10.69 | 0.060 | N | 0 | 0.06 | N | 0.030 | 0.04 |
| 2011-09-06 | N | 0 | 0.096 | N | 20.50 | 0.345 | N | 7.45E+08 | 0.326 | N | | 0.403 | N | 10.38 | 0.059 | N | 2 | 0.059 | N | 0.031 | 0.039 |
| 2011-09-07 | N | 0 | 0.094 | N | 18.18 | 0.338 | N | 1.40E+09 | 0.320 | N | | 0.395 | N | 10.93 | 0.058 | N | 1 | 0.058 | N | 0.030 | 0.038 |
| 2011-09-08 | N | 0 | 0.092 | Y | 15.87 | 0.350 | Y | 9.15E+08 | 0.314 | N | | 0.388 | N | 10.94 | 0.057 | N | 1 | 0.057 | N | 0.030 | 0.037 |
| 2011-09-09 | N | 0 | 0.09 | Y | 17.73 | 0.362 | N | 1.60E+09 | 0.308 | N | | 0.381 | N | 11.4 | 0.056 | N | 0 | 0.056 | N | 0.029 | 0.036 |
| 2011-09-10 | Y | 1 | 0.106 | Y | 21.18 | 0.373 | Y | 1.88E+09 | 0.303 | N | | 0.374 | N | 11.27 | 0.055 | N | 0 | 0.055 | N | 0.034 | 0.035 |
| 2011-09-11 | N | 0 | 0.104 | Y | 19.77 | 0.366 | N | 6.27E+08 | 0.298 | N | | 0.367 | N | 11.25 | 0.054 | N | 0 | 0.054 | N | 0.032 | 0.034 |
| 2011-09-12 | Y | 3 | 0.119 | Y | 19.69 | 0.360 | Y | 3.87E+09 | 0.293 | N | | 0.361 | N | 11.19 | 0.053 | N | 0 | 0.053 | N | 0.034 | 0.033 |
| 2011-09-13 | N | 1 | 0.117 | N | 22.36 | 0.354 | Y | 2.34E+09 | 0.288 | N | | 0.355 | N | 11.55 | 0.052 | N | 0 | 0.052 | N | 0.033 | 0.032 |
| 2011-09-14 | N | 0 | 0.115 | N | 20.04 | 0.348 | N | 7.86E+08 | 0.283 | N | | 0.349 | N | 11.4 | 0.051 | N | 0 | 0.051 | N | 0.038 | 0.031 |
| 2011-09-15 | N | 0 | 0.113 | N | 17.08 | 0.342 | N | 9.42E+08 | 0.278 | Y | | 0.360 | N | 11.83 | 0.050 | N | 0 | 0.05 | N | 0.034 | 0.03 |
| 2011-09-16 | N | 0 | 0.111 | N | 16.50 | 0.353 | N | 4.77E+08 | 0.274 | N | | 0.354 | N | 12.07 | 0.049 | N | 0 | 0.049 | N | 0.036 | 0.03 |
| 2011-09-17 | N | 0 | 0.109 | Y | 19.98 | 0.363 | N | 5.50E+08 | 0.270 | N | | 0.348 | N | 12.59 | 0.048 | N | 0 | 0.048 | N | 0.038 | 0.03 |
| 2011-09-18 | N | 0 | 0.107 | Y | 24.37 | 0.373 | Y | 1.08E+09 | 0.266 | N | | 0.343 | N | 12.76 | 0.047 | N | 0 | 0.047 | N | 0.038 | 0.03 |
| 2011-09-19 | N | 0 | 0.105 | Y | 25.80 | 0.383 | Y | 5.71E+08 | 0.262 | N | | 0.338 | N | 12.31 | 0.046 | N | 0 | 0.046 | Y | 0.042 | 0.045 |
| 2011-09-20 | Y | 8 | 0.119 | Y | 28.77 | 0.392 | Y | 8.51E+09 | 0.258 | N | | 0.333 | N | 12.57 | 0.045 | N | 0 | 0.045 | N | 0.041 | 0.044 |
| 2011-09-21 | N | 2 | 0.117 | Y | 26.78 | 0.386 | N | 1.69E+09 | 0.254 | N | | 0.328 | N | 12.53 | 0.044 | N | 1 | 0.044 | N | 0.041 | 0.043 |
| 2011-09-22 | N | 1 | 0.115 | N | 27.07 | 0.380 | N | 3.76E+09 | 0.250 | N | | 0.323 | N | 12.98 | 0.043 | N | 0 | 0.043 | Y | 0.046 | 0.057 |
| 2011-09-23 | Y | 6 | 0.128 | N | 24.97 | 0.374 | Y | 9.38E+09 | 0.246 | N | | 0.318 | N | 13.37 | 0.042 | N | 0 | 0.042 | N | 0.046 | 0.056 |
| 2011-09-24 | N | 2 | 0.126 | N | 23.50 | 0.369 | Y | 1.19E+10 | 0.257 | N | | 0.313 | N | 13.93 | 0.041 | N | 0 | 0.041 | N | 0.042 | 0.055 |
| 2011-09-25 | N | 2 | 0.124 | N | 15.41 | 0.364 | N | 3.57E+09 | 0.267 | N | | 0.309 | N | 12.82 | 0.040 | N | 0 | 0.04 | N | 0.047 | 0.054 |
| 2011-09-26 | Y | 11 | 0.136 | Y | 13.71 | 0.373 | Y | 1.21E+10 | 0.277 | N | | 0.305 | N | 13.73 | 0.039 | N | 0 | 0.039 | N | 0.047 | 0.053 |
| 2011-09-27 | Y | 48 | 0.148 | Y | 24.66 | 0.382 | Y | 1.24E+11 | 0.287 | Y | | 0.315 | N | 15.28 | 0.038 | N | 0 | 0.038 | N | 0.048 | 0.052 |
| 2011-09-28 | Y | 66 | 0.16 | Y | 25.60 | 0.390 | Y | 1.17E+11 | 0.297 | Y | | 0.324 | N | 15.58 | 0.037 | N | 0 | 0.037 | N | 0.048 | 0.065 |
| 2011-09-29 | Y | 49 | 0.171 | Y | 30.93 | 0.398 | Y | 1.57E+11 | 0.306 | Y | | 0.333 | Y | 16.12 | 0.050 | N | 0 | 0.037 | N | 0.050 | 0.064 |
| 2011-09-30 | N | 21 | 0.169 | N | 18.89 | 0.393 | Y | 3.21E+10 | 0.315 | Y | | 0.342 | N | 15.19 | 0.049 | N | 0 | 0.037 | N | 0.050 | 0.063 |
| 2011-10-01 | N | 27 | 0.167 | N | 15.67 | 0.388 | Y | 8.91E+10 | 0.324 | N | | 0.351 | N | 14.61 | 0.048 | N | 0 | 0.037 | N | 0.052 | 0.062 |
| 2011-10-02 | Y | 34 | 0.178 | N | 19.26 | 0.383 | N | 6.85E+10 | 0.333 | N | | 0.346 | N | 14.24 | 0.047 | N | 0 | 0.037 | N | 0.047 | 0.061 |
| 2011-10-03 | Y | 44 | 0.188 | N | 19.69 | 0.378 | N | 9.37E+10 | 0.341 | Y | | 0.354 | N | 14.58 | 0.046 | N | 0 | 0.037 | Y | 0.044 | 0.073 |
| 2011-10-04 | N | 17 | 0.186 | Y | 16.70 | 0.386 | N | 4.76E+10 | 0.349 | Y | | 0.362 | N | 14.82 | 0.045 | N | 0 | 0.037 | N | 0.048 | 0.072 |
| 2011-10-05 | N | 6 | 0.184 | Y | 19.74 | 0.394 | N | 2.63E+10 | 0.345 | Y | | 0.370 | N | 14.97 | 0.044 | N | 0 | 0.037 | N | 0.053 | 0.071 |
| 2011-10-06 | Y | 34 | 0.194 | Y | 25.24 | 0.401 | Y | 3.99E+10 | 0.341 | Y | | 0.378 | N | 14.39 | 0.043 | N | 0 | 0.037 | Y | 0.055 | 0.082 |
| 2011-10-07 | Y | 27 | 0.204 | Y | 43.17 | 0.408 | Y | 1.18E+11 | 0.349 | Y | | 0.385 | N | 13.56 | 0.042 | N | 0 | 0.037 | Y | 0.059 | 0.093 |
| 2011-10-08 | Y | 12 | 0.202 | Y | 46.39 | 0.415 | Y | 2.11E+11 | 0.357 | Y | | 0.392 | N | 12.25 | 0.050 | Y | 7 | 0.048 | Y | 0.055 | 0.104 |
| 2011-10-09 | N | 3 | 0.2 | Y | 25.62 | 0.422 | Y | 1.95E+10 | 0.365 | Y | | 0.399 | Y | 6.39 | 0.064 | N | 34 | 0.059 | N | 0.054 | 0.103 |
| 2011-10-10 | N | 2 | 0.198 | Y | 239.33 | 0.429 | N | 2.00E+09 | 0.361 | N | | 0.394 | Y | 11.55 | 0.075 | Y | 4 | 0.07 | N | 0.058 | 0.102 |





Figure 1

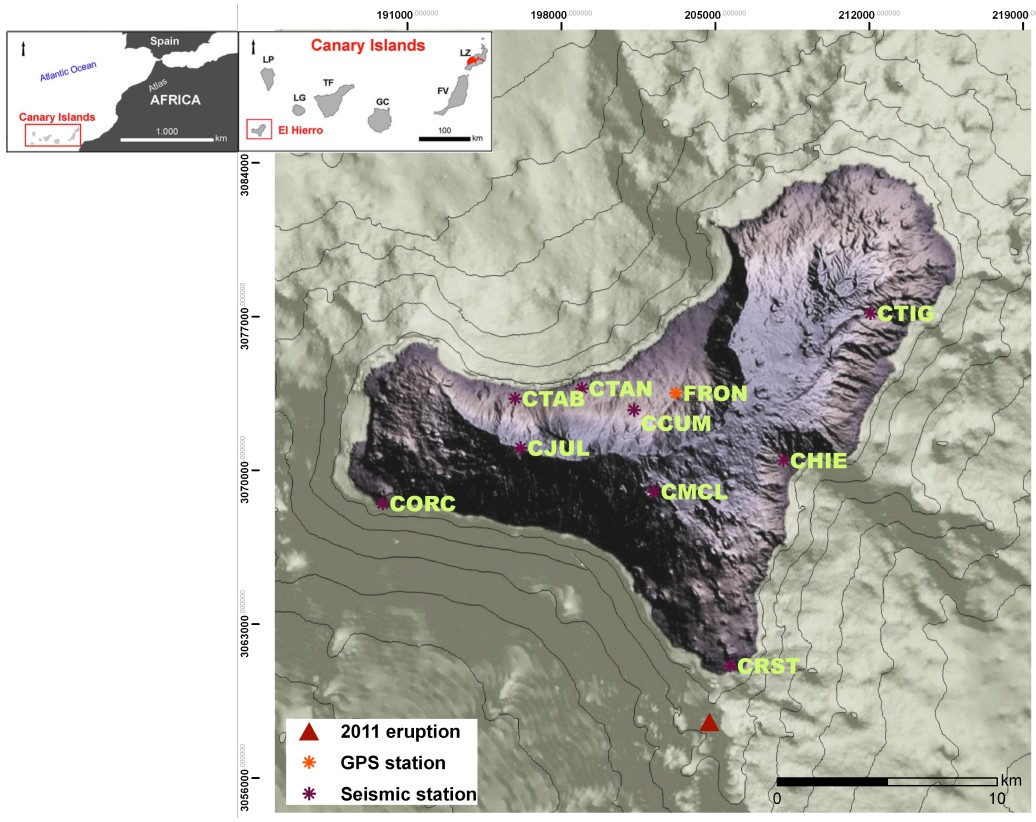





Figure 2

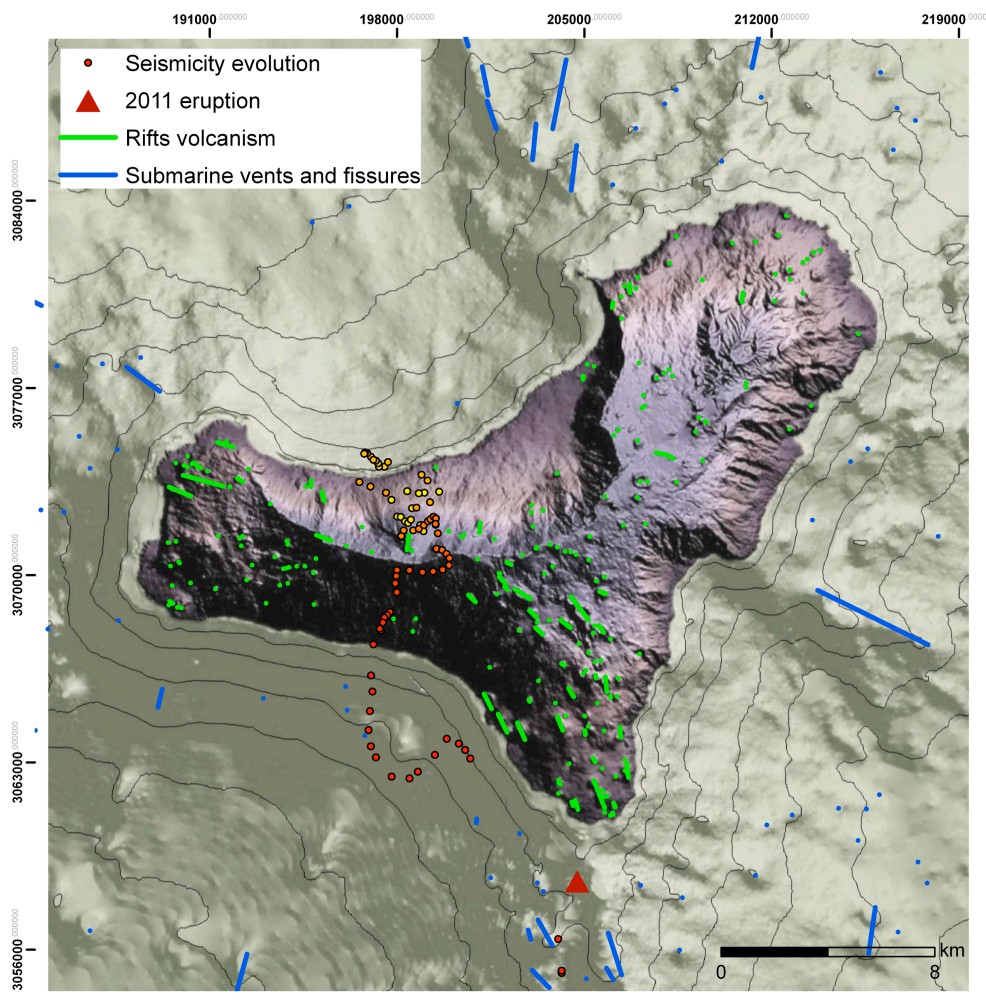





Figure 3

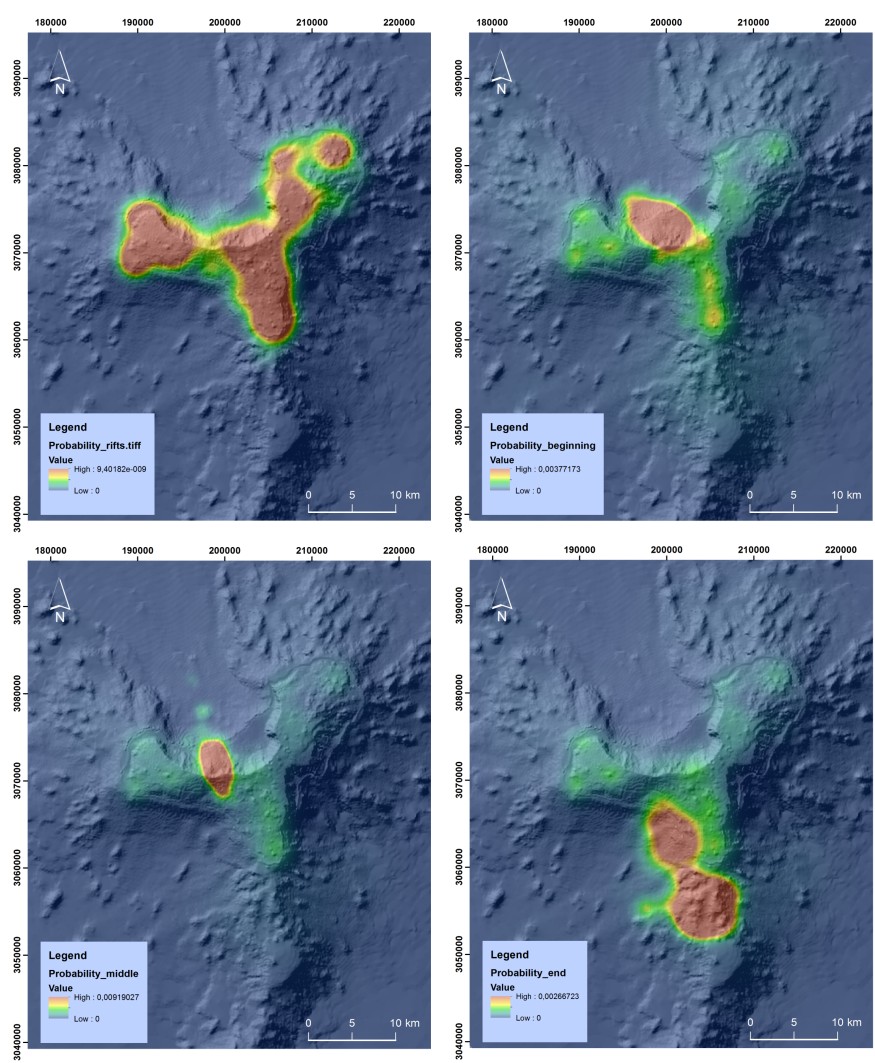





Figure 4

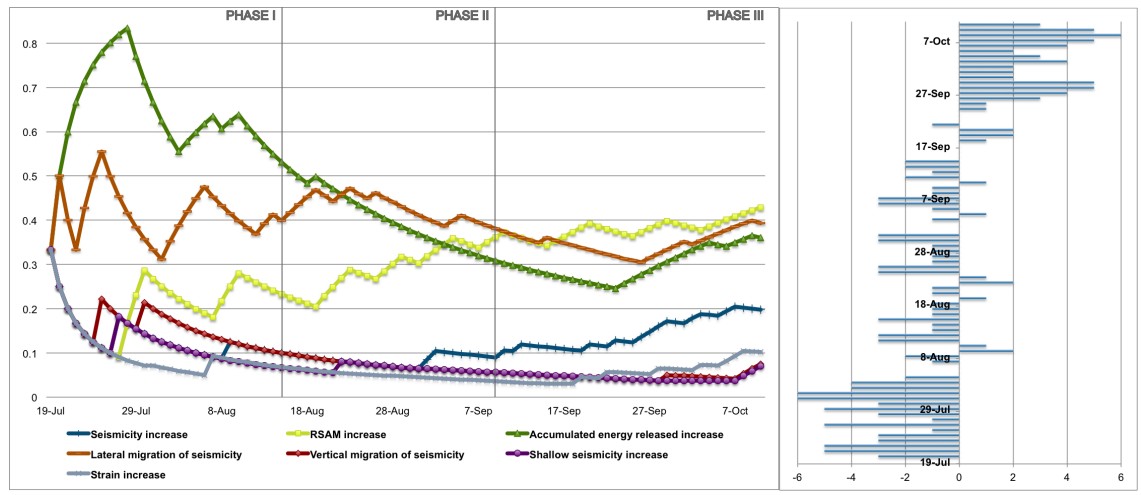



Figure 5

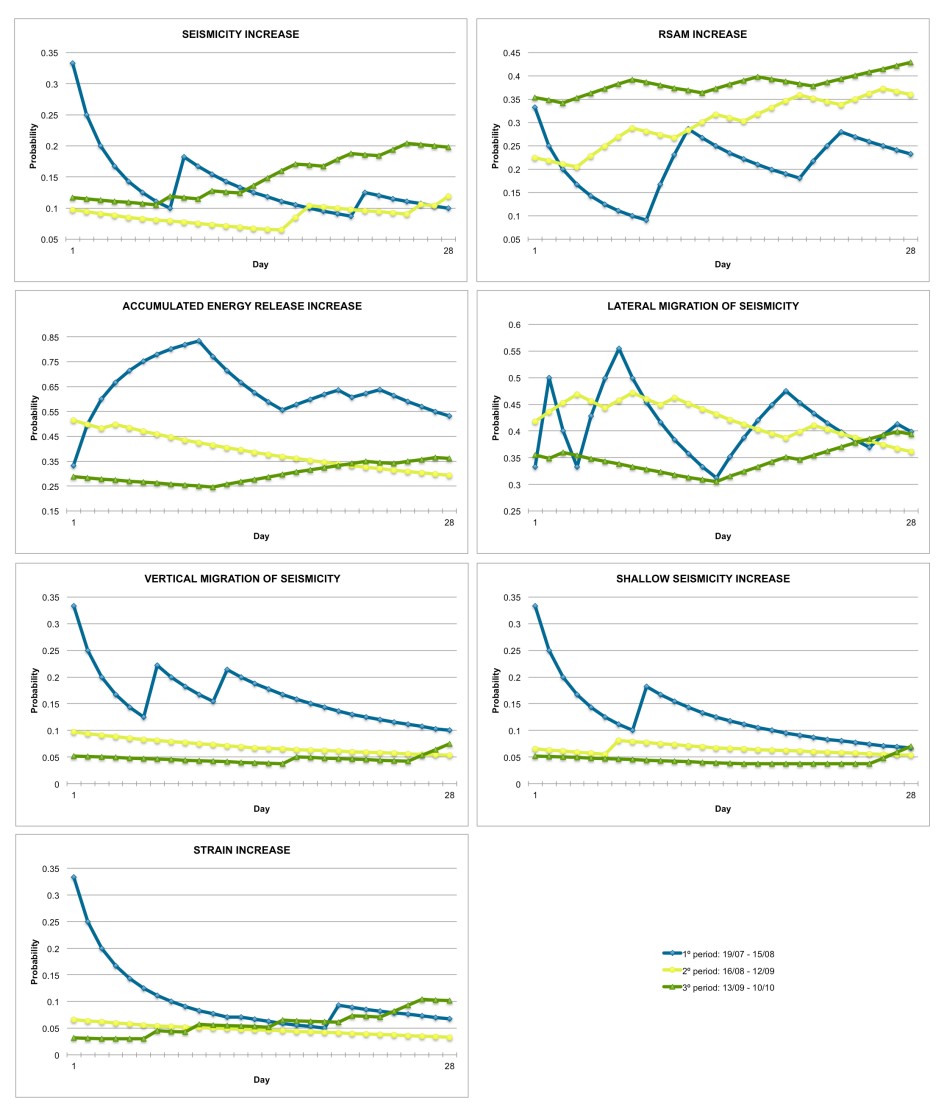