# Peer review of "A retrospective study of the pre-eruptive unrest on El Hierro (Canary Islands): implications of seismicity and deformation in the short-term volcanic hazard assessment"

_Natural Hazards and Earth System Sciences, 2017_

## Referee Comment (RC1) · Anonymous Referee #1 · 4 Sep 2017

This manuscript investigates the probability distribution of the location of the next vent opening and its temporal changes from seismic and deforamtion data. The authors look to apply the method to an assessment of volcanic activity in a real-time basis. This work is very well motivated because it is practically very important to forecast what happens next during a volcanic unrest in a real-time basis. I cannot really evaluate whether the methods the authors propose work well mainly because of the lack of information. In particular, the manuscript lacks a description of observed data which can be compared with the deduced probability distribution. Also the manuscript lacks a

description of methodological details. I understand that the they are described in previous publications but some descriptions are necessary so that a reader can reproduce what the authors got. With this, I find that this manuscript requires a substantial revision before reaching the publication quality. Please find my comments as enumerated below.

1. Section 2 overviews the methodology which mainly consists of QVAST and ST-HASSET. I followed a rough overview of what they do but did not understand mathematical details until I consult references cited in the manuscript. I would like the authors to show more mathematical datails of their methods, although a full description is not necessary.

2. Section 4.1 describes a probability distribution of future eruptive vents. I understand that it is deduced from eruptive vents and fissures, dikes, faults, fumaroles, and the stress field (lines 185-186) but do not understand how the authors made use of these information. I would like the authors to add some quantitative descriptions. possibly with figures.

3. Table 1 describes temporal evolution of various parameters which are required to assess the evolution of the state of the volcano. Making a table is a good idea but making a plot as well improve the visibility of the data. Also the authors need to show us how the "probability" in the table is derived from the "value". In addition, the authors need to describe how "Y/N/na" for each monitoring component is defined.

4. Figure 3 shows a temporal evolution of vent opening probability but I cannot evaluate whether it works because the actual location of vent opening is not shown in the figure.

5. Figure 4 shows a temporal evolution of probability associated with temporal changes of monitoring parameters, but similarly, I cannot evaluate if it works well because the observed data is not shown here.

6. Line 279: How can the "smoothing factors" be defined? Use equatinos.

[Figure]

7. Line 283: Show us what LSCV is more quantitatively.

8. Line 286: Show us what Silverman's Rule of Thumb more quantitatively.

9. Line 294 following weights: Seismic events, onshore vents and fissures, and off-shore vents and fissures have different units so how to define the weight is not straight-forward. I would like the authors to describe more quantitatively how to define the weight.

10. I find some awkward presentations in the manuscript. For example, a sentence from line 363 to 367 is too long and can be divided into two or more sentences for more clarification. I find some other awkward sentences but do not point out all. I would like the authors to take time to examine and rerun the presentation.

---

## Author Comment (AC1) · 14 Feb 2018

Dear NHESS editor, We would like to thank the referee for her/his comments that have been useful to improve some aspects of our manuscript.

In general, we consider that there is a lot of data already published and available to apply the methodology presented in this work. Please, see the IGN web page, http://www.ign.es/web/ign/portal/vlc-serie-hierro, and the publications cited in the manuscript. Also, with respect to the method applied, its description has been published in the previous paper Bartolini et al. (2016). Here, our purpose is to show an example of its application with real data using a retrospective approach.

Here we reply point by point to the referee's comments:

1. We consider repetitive to reproduce again all the mathematical concepts already explained and developed in previous papers that we adequately refer to in this new one. Our choice is also taken with the aim to facilitate the flow of the reading.

2. We have modified Figure 2 and the figure caption to facilitate the understanding of the geological structure described in the paper. (Figure 2. Structural data of El Hierro (vents and fissure onshore and offshore, as in Becerril et al. 2013, 2014) and the evolution of the seismicity during the unrest period (average location of the seismic swarm).)

3. Table 1 is not designed to plot the data. We only show all data used in the analysis. Anyway, we have modified the Table 1 trying to make it more readable.

4. and 5. We have modified Figure 3 and jointly with changes in Table 1 we have improved data visualization. Also, we suggest to see the Supplementary material where the video presented also helps to understand the methodology applied.

6. See the reply in comment 1.

7. and 8. Added some explication in the manuscript:

[...] This tool was applied first to evaluate the smoothing parameters or bandwidths of the dataset analysed, then to evaluate the probability density functions for each dataset, and, finally, to calculate the final susceptibility map (Fig. 3) (see also Figure S1). The bandwidth is a free smoothing parameter included in the kernel function that we used to estimate the corresponding probability density functions and determines how probabilities are distributed in terms of the distance from the volcanic structures or vents (Martí and Felpeto, 2010; Bartolini et al., 2013). In the case of the rift volcanism and the submarine layers, we applied the Least Square Cross Validation Method

(LSCV) (Cappello et al., 2012; Bartolini et al., 2013) to obtain the bandwidth parameter, as it better represents the geometry of the vents distribution, NE-SW elongated (see Becerril et al. (2013)). To determine the influence of seismicity in the spatial analysis, we considered that the most representative result was that obtained using Silverman's Rule of Thumb for the optimal bandwidth (Silverman, 1986). In fact, the result obtained using this method allows describing the spatial seismicity swarm distribution for the entire period, avoiding to underestimate the influence area (located close to the epi-central points) and to overestimate the density estimation (high values of the density distribution caused by small bandwidth values). [...]

9. Modified the sentence in the manuscript:

[...] In the evaluation of the final susceptibility, weights were assigned based on expert opinion and on previously published work (Becerril et al., 2013, 2014), and by taking into account the average depth of the seismicity during the unrest episode. In detail, the relevance and reliability values (Table 3) (Martí and Felpeto, 2010) have been assigned as follow: relevance was given through an elicitation of expert judgment procedure (Aspinall, 2006) among the members of the Group of Volcanology of Barcelona (GVB-CSIC) and external collaborators; reliability was considered as maximum in all the datasets (value of 1). Specifically, up to 7 October we observed no significant variation in the shallow seismicity (Table 1). In this case, we assigned the following weights: 0.5 for seismic events, 0.3 for onshore vents and fissures, and 0.2 for offshore vents and fissures. In the final period (8–10 October), we considered the shallow earthquakes as a separate layer by assigning a different and more consistent weight as follows: 0.6 for shallow seismic events, 0.2 for the remaining seismic events, 0.1 for onshore vents and fissures, and 0.1 for offshore vents and fissures. [...]

10. Changed.

**Fig. 1.**

**Fig. 2.**

---

## Referee Comment (RC2) · Anonymous Referee #2 · 2 Apr 2018

This work describes the analysis of unrest indicators such as seismicity, RSAM, surface deformation and gas emissions at El Hierro during the crisis started in July 2011 and that culminated in the submarine eruption in October 2011. The analysis is conducted by using methods previously developed by the same authors (eg: QVAST, ST-HASSET). The analysis was performed a-posteriori. Results are very interesting since highlight the relative importance of the different parameters during the crisis, showing, in particular, the increase of the RSAM and the role of the other seismic parameter in defining the area with maximum probability of vent opening.

[Figure]

Minor corrections:

Line 165: "GPS station began to rotate to the North". Perhaps the authors mean that the GPS station translated towards North.

Line 215: "Data was grouped" -> "Data were grouped"

Lines 254-256: The authors consider as a new parameter the "slope of the inverse of the RSAM". This is not clear, since it could be interpreted either as the instantaneous derivative respect to time of the parameter (1/RSAM) [ie: d(1/RSAM)/dt] or the slope of (1/RSAM) taken in a larger time-window (eg: the slope of the line best-fitting 1/RSAM, etc.). Moreover, this new parameter is not reported in the figures.

Line 367: "eruptive phase". Do you mean "unrest phase"?

Line 439: It seems that the eruption began 11 October, in contrast with line 152 (10 October).

---

## Author Comment (AC2) · 5 Apr 2018

We would like to thank the Anonymus Referee #2 for her/his comments useful to improve some aspects of our manuscript.

We here reply point by point at the comments:

1. Line 165: "GPS station began to rotate to the North". Perhaps the authors mean that the GPS station translated towards North.

Changed

2. Line 215: "Data was grouped" -> "Data were grouped"

Modified

3. Lines 254-256: The authors consider as a new parameter the "slope of the inverse of the RSAM". This is not clear, since it could be interpreted either as the instantaneous derivative respect to time of the parameter (1/RSAM) [ie: d(1/RSAM)/dt] or the slope of (1/RSAM) taken in a larger time-window (eg: the slope of the line best-fitting 1/RSAM, etc.). Moreover, this new parameter is not reported in the figures.

We consider the slope of the line best-fitting (1/RSAM). Modified in Table 1 and in the Supplementary material the RSAM with (1/RSAM)

4. Line 367: "eruptive phase". Do you mean "unrest phase"?

We refer to the eruptive phase, because after previous unrest phases the system is getting ready to erupt

5. Line 439: It seems that the eruption began 11 October, in contrast with line 152 (10 October).

Modified
* * *